# HE4 in the Diagnostic Approach of Endometrial Cancer in Patients with Postmenopausal Bleeding, the METRODEC Protocol: Protocol for a Multicenter Prospective Study

**DOI:** 10.3390/diagnostics11071274

**Published:** 2021-07-15

**Authors:** Manon Degez, Hélène Caillon, Anne Chauviré-Drouard, Maxime Leroy, David Lair, Norbert Winer, Thibault Thubert, Pauline Le Floch, Valérie Desroys du Roure, Mélanie Randet, Guillaume Ducarme, Vincent Dochez

**Affiliations:** 1Service de Gynécologie-Obstétrique, CHU de Nantes, 44000 Nantes, France; manondegez@msn.com (M.D.); norbert.winer@chu-nantes.fr (N.W.); Thibault.thubert@chu-nantes.fr (T.T.); 2Service de Biochimie, CHU de Nantes, 44000 Nantes, France; helene.caillon@chu-nantes.fr; 3Centre d’Investigation Clinique CIC 1413, INSERM, CHU de Nantes, 44000 Nantes, France; anne.drouard@chu-nantes.fr; 4Plateforme de Biométries et Biostatistiques, CHU de Nantes, 44000 Nantes, France; maxime.leroy@chu-nantes.fr; 5Département Promotion, Direction de la Recherche, CHU de Nantes, 44000 Nantes, France; david.lair@chu-nantes.fr; 6Unité de Recherche Clinique, CH de Saint-Nazaire, 44600 Saint-Nazaire, France; p.lefloch@ch-saintnazaire.fr; 7Unité de Recherche Clinique, CH Départemental Vendée, 85000 La Roche sur Yon, France; valerie.desroysduroure@chd-vendee.fr; 8Service de Gynécologie-Obstétrique, CH de Saint-Nazaire, 44600 Saint-Nazaire, France; m.randet@ch-saintnazaire.fr; 9Service de Gynécologie-Obstétrique, CH Départemental Vendée, 85000 La Roche sur Yon, France; guillaume.ducarme@chd-vendee.fr

**Keywords:** endometrial cancer, human epididymis protein 4, HE4, CA125, postmenopausal bleeding, hysteroscopy

## Abstract

Background: Endometrial cancer is the most common pelvic gynecological cancer in France. The most frequent symptom is post-menopausal bleeding and is one of the primary reasons for consultation in gynecological emergencies. The treatment is very codified and consists of a surgical intervention for anatomopathological analysis. The latter is frequently reassuring. These interventions are often performed in mild situations and there is currently no element to be sufficiently reassuring to avoid surgery. This study aims to explore the sensitivity of an innovative marker: Human Epididymis 4 (HE4) in the diagnosis approach of endometrial cancer in case of postmenopausal bleedings. Methods: This is a prospective multicenter diagnostic study with three centers involved. Inclusion criteria are any patient with post-menopausal bleeding who is to undergo hysteroscopy, endometrial biopsy, or endometrial resection. In accordance with the recommendations for the management of post-menopausal bleedings, the medical conduct consists of performing a clinical examination, an ultrasound and, in general, even in case of paraclinical examination reassuring, an anatomopathological analysis. This pathological analysis can be obtained in several ways: biopsy, hysteroscopy-curettage (which is the most frequently performed surgery), and hysterectomy. Our protocol consists of taking a blood sample from each woman who will undergo one of the interventions mentioned above. The dosage of HE4 and CA125 requires the withdrawal of an additional heparinized tube during the preoperative assessment usually performed. This research is therefore classified as non-interventional. The primary outcome is to evaluate the sensitivity of the HE4 marker in patients with postmenopausal bleeding in the diagnosis of endometrial cancer. The secondary outcomes are other parameters (specificity, VPP, VPN) of HE4, Evaluating the diagnostic capabilities of the CA125 marker alone and associated with HE4, as well as those of the REM and REM-B algorithms. We aim to include 100 patients over a period of one year in three centers. Discussion: As of now, there is no biological marker used in routine practice in the diagnosis of endometrial cancer. The ultimate goal of HE4 in endometrial cancer is to avoid surgery for those who are identified as non-sick. This study is the precursor of others for use in routine practice, HE4 would represent a great help to diagnosis if our study demonstrates it as reliable in the management of these patients and avoid many unnecessary and risky surgeries.

## 1. Introduction

Endometrial cancer is the most common gynecological pelvic cancer with 390,000 cases and 90,000 deaths worldwide in 2018 [1,2,3] In France, it is the 5th most common cancer affecting women, with more than 7000 new cases per year (compared with around 4500 cases per year for ovarian cancer) [4]. The symptoms most frequently found are postmenopausal bleeding (PMB) and represent one of the first reasons for consultation in gynecological emergencies. Relative survival at 5 years is 76% overall. For a localized stage (almost 70% of diagnoses), it increases to 95% [5].

In 2010, the recommendations of the HAS and the National Cancer Institute codify the management of post-menopausal bleeding, which consists of a clinical examination, an ultrasound scan and, in general, even in the case of a reassuring paraclinical examination, an anatomopathological analysis (Gold standard) [5]. Recent guidelines published in 2021 [6] advise that “In case of recurrent abnormal uterine bleeding or when the endometrium thickness is greater than 4 mm in a postmenopausal woman, additional uterine investigations are recommended”.

This anatomopathological analysis can be obtained in several ways:-By endometrial biopsy (which is only of value if it finds a positive result in favor of a malignant lesion).-By hysteroscopy with endometrial biopsy or endometrial resection which is the most frequently performed surgery.-By hysterectomy from the outset in the case of very incapacitating metrorrhagia (less used method).

In the population of patients with PMB requiring surgical exploration for diagnosis, the anatomopathological results are often reassuring, as the histological analysis is benign. These operations could therefore have been avoided, especially as these patients often have numerous comorbidities (high age, overweight, hypertension, etc.), and these operations are therefore risker. In a meta-analysis published in 2018, it was found that the prevalence of PMB among women with endometrial cancer was 91% (95% CI, 87–93%), irrespective of tumor stage. Nonetheless, the risk of endometrial cancer among women with PMB was 9% (95% CI, 8–11%) [7].

According to a retrospective study carried out at the Nantes University Hospital over two years from 2017 to 2018, 100 hysteroscopies were performed for post-menopausal bleeding. Among these 100 surgical procedures, 22 endometrial cancers were found. This means that 78 procedures were performed for metrorrhagia of benign origin and therefore a majority of procedures could have been avoided.

Some hysteroscopies may have therapeutic value even in benign conditions, but many situations could be managed medically or monitored simply as a first step. However, this type of strategy can only be envisaged if there is a blood marker analysis that would place the patient in a low-risk category. At present, there is no biological marker used in current practice for the diagnosis of endometrial cancer. The appearance of a new tumor marker could therefore be useful in the management of these patients and avoid many unnecessary and risky operations.

HE4 (Human Epididymis 4), also known as WAP four-disulfide core domain protein 2 (WFDC2), is a blood biomarker previously evaluated in the literature for ovarian cancer screening and used as an aid to surgical management decisions. It was originally identified in the distal epididymis and plays a role in sperm maturation. Its recent use is due to the fact that it is normally expressed in various tissues including ovarian but overexpressed in ovarian cancer particularly in endometrioid types [8]. According to the new National Cancer Institute in France (INCa), recommendations established in 2018 “Serum HE4 determination is recommended for the diagnosis of an indeterminate ovarian mass on imaging (Grade A)” [9,10].

Furthermore, while the treatment of ovarian cancers is complex due to the multiple histological types encountered, endometrial cancers are more monomorphic. Indeed, epithelial tumors represent more than 90% of endometrial cancers. The remaining 10% are made up of numerous histological types, each much rarer, such as sarcomas. The interpretation of HE4 in this indication therefore seems much easier and can be extrapolated [11].

Although few data are available at present, some studies have recently demonstrated the interest of Human Epididymis 4 (HE4) in the detection of endometrial cancer with a high sensitivity and a specificity close to 100% for some authors. In a recent literature review published by our team in December 2020, we conclude that HE4 alone, or in association with CA125, would be a relevant tool in the diagnosis of endometrial cancer. Moreover, in the same review, HE4 seems to be associated with myometrial invasion, International Federation of Gynecology and Obstetrics (FIGO) stage, or lymph node invasion, but also as a predictive factor of recurrence [12,13,14].

Yilmaz et al. established in 2017 a clear correlation between elevated HE4 values and endometrial cancer pathology [14]. Among patients with abnormal metrorrhagia, those in the cancer group had significantly higher HE4 values 91.4 pmol/L versus 46.2 pmol/L (*p* < 0.001). They found a sensitivity of 72.7% and a specificity of 84.4% in detecting endometrial cancer in metrorrhagia [12].

However, the HE4 marker can be increased in other benign situations such as age, smoking, renal failure, tubo-ovarian abscesses, or malignant in pancreatic adenocarcinoma, borderline ovarian tumors. We therefore try to take these confounding factors into account in order to better evaluate and interpret the variations of the HE4 marker [15].

The aim of our study is therefore to evaluate the value of the HE4 marker in patients with postmenopausal bleeding for the diagnosis of endometrial cancer. In our study, we chose to evaluate the sensitivity of the HE4 marker as a priority. In practice, the ultimate goal of HE4 in endometrial cancer would be to avoid surgical intervention in those identified as non-diseased, but the essential element is not to ignore a pathological situation and therefore to obtain a high sensitivity with very few false negatives as a first objective.

## 2. Methods and Design

### 2.1. Trial Design

This is a non-interventional prospective multicenter diagnostic study.

### 2.2. Participating Centers

The three participating centers are the Nantes university hospital Center (UHC) (Nantes, 44, France), the Departmental Hospital Center of Vendée (La Roche sur Yon, 85, France), and the Saint Nazaire Hospital Center (Saint Nazaire, 44, France).

### 2.3. Eligibility Criteria

Patients are recruited from among the women with post-menopausal bleeding who is to undergo hysteroscopy with endometrial biopsy or endometrial resection or hysterectomy.

### 2.4. Inclusion Criteria

-Patient with postmenopausal bleeding;-Patient requiring hysteroscopy or hysterectomy;-No objection from the patient to participate in the study.

### 2.5. Exclusion Criteria

-Non-menopausal patient;-Patient under guardianship, curatorship, or deprived of her freedom;-Patient with proven metastases on imaging;-Patient with a macroscopically suspicious cervix;-Patient presenting an ovarian cyst or an associated adnexal pathology;-Patient contraindicated for surgical treatment (therefore not eligible for pathological analysis);-Patient who has already been treated with hormone therapy for breast cancer;-Patient who has already had surgery for this pathology, with a contributing anatomopathological result (we therefore include patients who would not benefit from an operative hysteroscopy after performing an endometrial biopsy that does not allow a positive result).

### 2.6. Materials

This is a prospective multicenter non-interventional diagnostic study conducted in three centers with an expected recruitment of 100 patients over 12 months.

Any patient presenting post-menopausal bleeding who is to undergo hysteroscopy, for endometrial biopsy or endometrial resection, has an additional blood tube taken during her pre-operative blood work-up for the determination of the serum markers HE4 and CA125, after having obtained her non-opposition to participate in the study.

During the visit where the surgeon confirms the operative indication, the patient is informed of the study and the modalities of inclusion. A letter of information is given and the patient is asked to give her consent by non-opposition. During this inclusion visit, the various inclusion and non-inclusion criteria is checked.

On inclusion, a code is provided for the patient via the electronic CRF, which allows the tube and the sampling form to be labeled.

To facilitate sampling, the pre-operative work-up is performed following the inclusion visit at the inclusion center. An additional tube is collected to perform the CA125 and HE4 assay.

The blood tube collected for the research is sent to the biochemistry laboratory of the Nantes University Hospital for a centralized analysis.

Blood is collected in a standard heparinized vial. Samples are be sent to a central laboratory unit (biochemistry laboratory at the Nantes university hospital), centrifuged and are carried out gradually. − Plasma CA125 and HE4 concentrations are determined by a run in single measurements using an electrochemiluminescence Elecsys immunoassay (ECLIA) on a Roche Diagnostics Cobas Pro module^®^ E801 analyzer (Roche Diagnostics, Mannheim, Germany). This analysis is done as we go along and the results are not transmitted to the investigator.

At D0, the day of surgery, the operative report is retrieved as well as the quality-of-life questionnaires SF12 and PGI-I and the acceptability questionnaire completed before surgery.

At 1 month after surgery, the anatomopathological results are retrieved as well as the value of the tumor.

### 2.7. Outcome Measures

#### 2.7.1. Primary Outcome

The sensitivity of the HE4 marker is evaluated in patients with postmenopausal bleeding in the diagnosis of endometrial cancer.

#### 2.7.2. Secondary Outcomes

-Other diagnostic parameters (specificity, positive predictive value (PPV), negative predictive value (NPV)) and the Area Under the Curve (AUC) of HE4 are assessed.-The optimal threshold of HE4 for the diagnosis of endometrial cancer are established with use of a Receiver Operating characteristic (ROC) curve.-The diagnostic capabilities of CA125 alone and in combination with HE4 are evaluated, as well as the Risk of Endometrial Malignancy (REM) and Risk of Endometrial Malignancy associated with Body Mass Index (REM-B) algorithms for the diagnosis of endometrial cancer.-(Estimation of sensitivity, specificity, PPV and NPV in the diagnosis of endometrial cancer for the CA125 biomarker and the REM and REM-B algorithms).-The existence of thresholds for HE4 and/or CA125 markers predictive of disease severity are established (FIGO stage).-We reassess the pathological threshold value of endometrial thickness on ultrasound.-The relationship between endometrial thickness on ultrasound and HE4 and CA125 marker values are assessed.-The diagnostic capabilities of HE4 and CA125 in subgroups of smoking patients and patients with renal failure are evaluated.-We identify potential confounding factors associated with the value of the HE4 marker like treatments, comorbidities (renal insufficiency, high Body Mass Index (BMI)), and other criteria that are collected in the medical files.

Variables measured at inclusion that may influence the value of the HE4 marker are analyzed.

### 2.8. Sample Size

In a retrospective study of hysteroscopies performed for postmenopausal metrorrhagia in 2017 and 2018 at the Nantes University Hospital, 22 endometrial cancers were found in 100 patients with postmenopausal metrorrhagia who underwent hysteroscopy for endometrial biopsy or endometrial resection. Assuming that 80% of the patients are included in this study, the inclusion capacity of the Nantes center is 40 patients over 1 year, i.e., approximately 3 to 4 patients per month. For the other two hospitals, a recruitment of 2 to 3 patients per month is envisaged. This brings the theoretical number of patients to more than 100 per year, which makes the study feasible in terms of recruitment.

### 2.9. Statistical Methods

A descriptive analysis is carried out on all the variables collected. The analyses are be carried out using R software version 4.0. The diagnostic capacities of the markers are evaluated by estimating the ROC curve and calculating the positive (PPV) and negative (NPV) predictive values. The performance of the diagnostic tests is assessed by comparing the areas under the ROC curve of these markers by the non-parametric DeLong and Clarke–Pearson test.

The relationship between HE4 values and the different FIGO stages is established by using linear mixed models with a random effect on the center.

## 3. Discussion

HE4 currently seems to be the marker that offers the most promise in terms of diagnosis, prognosis, and recurrence monitoring, either alone or in combination with CA125 [12,13]. The use of algorithms such as REM and REM-B also seems to be an interesting tool. Our ultimate goal is to avoid surgery and to propose simple surveillance for those identified as not at risk of cancer using the HE4 marker.

To date, most studies have evaluated this marker in the general population, and very few in a target population (symptomatic and post-menopausal patients). Furthermore, no French study has been conducted on the HE4 marker and endometrial cancer.

In 2019, the team of Jinping Li et al. [14] published a meta-analysis evaluating the markers HE4 and CA125. Of the 12 studies included from 1991 to 2013, five were in a Caucasian population and seven in a Chinese population, with patients with cancer or benign conditions. In total, 1106 patients had endometrial cancer and 1480 had benign disease. For the HE4 marker, a sensitivity of 0.71 (95% CI 0.56–0.82) and a specificity of 0.87 (95%CI 0.80–0.92) were found, compared with the CA125 marker, sensitivity 0.35 (95% CI 0.25–0.46), and specificity 0.83 (95% CI 0.71–0.91). Subgroup analyses showed an even higher sensitivity of 0.86 (95% CI 0.64–0.96) and a good specificity of 0.84 (95% CI 0.68–0.92) for the Caucasian population. However, this meta-analysis is very heterogeneous, including both prospective and retrospective studies, with a variable population.

In the study by Yilmaz et al. [15], a correlation was found between HE4 levels and disease severity. A link was found between HE4 levels and FIGO stage (RR = 0.482, *p* < 0.001), between endometrial thickness and HE4 levels, between lymph node invasion and HE4 levels (RR = 0.387; *p* < 0.001) and with tumor size (RR = 0.386, *p* < 0.001). Thus, according to lymph node invasion, the more advanced the pathology, the higher the HE4 rate [16,17]. Again in 2017, Dewan et al. [18] found the same results, with a link between the increase in the value of the HE4 marker and the severity of the disease, the higher the level, the more myometrial and/or lymph node invasion is found. Finally, in a Danish study carried out from 2009 to 2012, an association was also found between the importance of the FIGO stage, myometrial invasion, or lymph node invasion (*p* < 0.005) [19].

The aim of our study is therefore to evaluate the value of the HE4 marker in patients with postmenopausal for the diagnosis of endometrial cancer.

In our study, we chose to focus on the sensitivity of the HE4 marker. It is always difficult to prioritize sensitivity or specificity and the aim is that both values are optimal.

However, in our study, the population studied is a high-risk population (postmenopausal bleeding), and we have no other reliable tests to avoid surgery. Obtaining a high sensitivity implies the detection of a really affected patient, whereas a high specificity identifies a population without malignant lesions as not being affected.

Moreover, current medical developments favor diagnostic approaches that are less and less invasive and more personalized, adapted to the patient for a better benefit–risk. HE4 could then be used to guide the gynecologist in the diagnostic approach to post-menopausal bleeding, with the aim of reducing unnecessary interventions, by proposing simple monitoring for patients at low risk. This categorization can only be done through the analysis of a reliable marker. At present, HE4 seems to be the marker with the most potential in terms of diagnosis either alone or in combination with CA125. Clinical data are still needed to recommend the prescription of HE4 in this indication and a French clinical study involving HE4 has yet to be conducted. Our METRODEC study is part of this innovative approach, aimed at optimizing the management of patients with post-menopausal bleeding. The results of our study, expected in September 2022, will hopefully indicate the relevance of the HE4 marker in this use.

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
