# Peer review of "HE4 in the Diagnostic Approach of Endometrial Cancer in Patients with Postmenopausal Bleeding, the METRODEC Protocol: Protocol for a Multicenter Prospective Study"

_diagnostics, 2021, doi:10.3390/diagnostics11071274_

Round 1

Reviewer 1 Report

The introduction should explain why it will be important to designate HE4, because after reading this introduction, there are further doubts. Hisatological verification is available anyway. The authors do not suggest that the level of the He4 marker will allow for the selection of the surgical method, e.g. laparoscopy, laparotomy. The inclusion and exclusion criteria need to be fine-tuned. I do not think there were included patients with dialysis, inflammation failure, liver, etc. where the marker levels would be elevated. In addition, the discussion requires improvement and a more in-depth discussion of the topic with other authors. Below are some examples of publications

Zamani, N.; Modares Gilani, M.; Mirmohammadkhani, M.; Sheikhhasani, S.; Mousavi, A.; Yousefi Sharami, S.R.; Akhavan, S.; Zamani, M.H.; Saffarieh, E. The Utility of CA125 and HE4 in Patients Suffering From Endometrial Cancer. International Journal of Women’s Health and Reproduction Sciences 2019, 8, 95–100, doi:10.15296/ijwhr.2020.14.

            Angioli, R.; Miranda, A.; Aloisi, A.; Montera, R.; Capriglione, S.; De Cicco Nardone, C.; Terranova, C.; Plotti, F. A Critical Review on HE4 Performance in Endometrial Cancer: Where Are We Now? Tumour Biol. 2014, 35, 881–887, doi:10.1007/s13277-013-1190-4.

            Angioli, R.; Plotti, F.; Capriglione, S.; Scaletta, G.; Dugo, N.; Aloisi, A.; Piccolo, C.L.; Del Vescovo, R.; Terranova, C.; Zobel, B.B. Preoperative Local Staging of Endometrial Cancer: The Challenge of Imaging Techniques and Serum Biomarkers. Arch. Gynecol. Obstet. 2016, 294, 1291–1298, doi:10.1007/s00404-016-4181-z.

            Angioli, R.; Plotti, F.; Capriglione, S.; Montera, R.; Damiani, P.; Ricciardi, R.; Aloisi, A.; Luvero, D.; Cafà, E.V.; Dugo, N.; et al. The Role of Novel Biomarker HE4 in Endometrial Cancer: A Case Control Prospective Study. Tumour Biol. 2013, 34, 571–576, doi:10.1007/s13277-012-0583-0.

 Cymbaluk-PÅ‚oska A, GarguliÅ„ska P, Bulsa M,Kwiatkowski S, Chudecka-GÅ‚az A.Michalczyk K. Can the Determination of HE4 and CA125 Markers Affect the Treatment of Patients with Endometrial Cancer? Diagnostic 202111(4):626. doi: 10.3390/diagnostics11040626.

Author Response

Dear Reviewers,

Thank you for your reading and your comments.

We deeply appreciate the time and suggestions of the reviewer and have responded to his comments. The manuscript has been modified in accordance with his suggestions.

We hope that these changes in the revised manuscript now make our paper suitable for publication.

Best regards,

Vincent Dochez

HE4 in the diagnostic approach of endometrial cancer in patients with postmenopausal bleeding, the METRODEC Protocol: Protocol for a multicenter prospective study

Response to reviewer 1:

-The introduction should explain why it will be important to designate HE4, because after reading this introduction, there are further doubts. Histological verification is available anyway.

Thank you for your analysis and reading of our protocol

As a response we suggest:

Page 3 Line 124-127

In practice, the ultimate goal of HE4 in endometrial cancer would be to avoid surgical intervention in those identified as non-diseased, but the essential element is not to ignore a pathological situation and therefore to obtain a high sensitivity with very few false negatives as a first objective.

-The authors do not suggest that the level of the He4 marker will allow for the selection of the surgical method, e.g. laparoscopy, laparotomy.

The HE4 marker will help the gynecologist to classify patients as being at high or low risk of endometrial cancer.

Thus, patients will either have a histological analysis, by the surgical technique wanted by the medical team, or the patients will be able to benefit from a simple follow-up if they are in a low risk category.

But the level of HE4 is not used to choose the surgical method.

-The inclusion and exclusion criteria need to be fine-tuned. I do not think there were included patients with dialysis, inflammation failure, liver, etc. where the marker levels would be elevated. 

Our study is a preliminary clinical trial and our aim is to focus on the sensitivity of HE4. We have few exclusion criteria as we want to study HE4 also in cases where there might be false positives and confounding factors. These are often dialysis and frail patients who might benefit from simple monitoring and avoid hysteroscopy.

In addition, the “Patient Protection Committee (CPP) of Ouest III” approved this protocol on 31/05/2021 as reference RC21_0166. The study will be conducted in compliance with the current approved version of the protocol. We have also obtained the authorization of the National Commission for Data Processing and Liberties (French CNIL). The METRODEC study has been listed in the ClinicalTrials database since April 30, 2021, as NCT04867109.

Thus, if our results are in favor of the relevance of HE4 in our indication, we would like to move towards a more targeted protocol, in particular like excluding this type of confounding factors, in order to carry out a more in-depth and more targeted study with medico-economic criteria.

-In addition, the discussion requires improvement and a more in-depth discussion of the topic with other authors. Below are some examples of publications

As a response we suggest:

-Page 6 Line 263-268: Moreover, current medical developments favor diagnostic approaches that are less and less invasive and more personalized, adapted to the patient for a better benefit-risk. HE4 could then be used to guide the gynaecologist in the diagnostic approach to post-menopausal bleeding, with the aim of reducing unnecessary interventions, by proposing simple monitoring for patients at low risk. This categorization can only be done through the analysis of a reliable marker.

-Page 6 line 271-274: Our METRODEC study is part of this innovative approach, aimed at optimizing the management of patients with post-menopausal bleeding. The results of our study, expected in September 2022, will hopefully indicate the relevance of the HE4 marker in this use

-Page 5 line 232-236: At present, HE4 seems to be the marker that offers the most promises in terms of diagnosis, prognosis and recurrence monitoring, either alone or in combination with CA125 (12-13). The use of algorithms such as REM and REM-B also seems to be an interesting tool. Our ultimate goal is to avoid surgery and to propose simple surveillance for those identified as not at risk of cancer using the HE4 marker.

Zamani, N.; Modares Gilani, M.; Mirmohammadkhani, M.; Sheikhhasani, S.; Mousavi, A.; Yousefi Sharami, S.R.; Akhavan, S.; Zamani, M.H.; Saffarieh, E. The Utility of CA125 and HE4 in Patients Suffering From Endometrial Cancer. International Journal of Women’s Health and Reproduction Sciences 2019, 8, 95–100, doi:10.15296/ijwhr.2020.14.

            Angioli, R.; Miranda, A.; Aloisi, A.; Montera, R.; Capriglione, S.; De Cicco Nardone, C.; Terranova, C.; Plotti, F. A Critical Review on HE4 Performance in Endometrial Cancer: Where Are We Now? Tumour Biol. 2014, 35, 881–887, doi:10.1007/s13277-013-1190-4.

            Angioli, R.; Plotti, F.; Capriglione, S.; Scaletta, G.; Dugo, N.; Aloisi, A.; Piccolo, C.L.; Del Vescovo, R.; Terranova, C.; Zobel, B.B. Preoperative Local Staging of Endometrial Cancer: The Challenge of Imaging Techniques and Serum Biomarkers. Arch. Gynecol. Obstet. 2016, 294, 1291–1298, doi:10.1007/s00404-016-4181-z.

            Angioli, R.; Plotti, F.; Capriglione, S.; Montera, R.; Damiani, P.; Ricciardi, R.; Aloisi, A.; Luvero, D.; Cafà, E.V.; Dugo, N.; et al. The Role of Novel Biomarker HE4 in Endometrial Cancer: A Case Control Prospective Study. Tumour Biol. 2013, 34, 571–576, doi:10.1007/s13277-012-0583-0.

 Cymbaluk-PÅ‚oska A, GarguliÅ„ska P, Bulsa M,Kwiatkowski S, Chudecka-GÅ‚az A.Michalczyk K. Can the Determination of HE4 and CA125 Markers Affect the Treatment of Patients with Endometrial Cancer? Diagnostic 202111(4):626. doi: 10.3390/diagnostics11040626.

Reviewer 2 Report

In developed countries, endometrial cancer is currently the most common malignant neoplasm of female genital organs.

Histological verification is a widely recognized diagnostic method aimed at the diagnosis of this disease. In order to avoid aggressive invasive diagnostics and unjustified surgical intervention, other methods of early diagnosis of this neoplasm are sought.

At present, there is no biological marker used in current practice for the diagnosis of endometrial cancer.

The aim of the study submitted for review was prepare a protocol for a multicenter prospective study to evaluate the value of the HE4 marker in  patients with postmenopausal bleeding for the diagnosis of endometrial cancer. In this study, the authors chose to evaluate the sensitivity of the HE4 marker as a priority.

The reviewed work fits perfectly into the current of research on this subject in leading oncology centers in the world. The proposed research methods do not raise any objections.

Author Response

Dear Reviewers,

Thank you for your reading and your comments.

We deeply appreciate the time and suggestions of the reviewer and have responded to his comments. The manuscript has been modified in accordance with his suggestions.

We hope that these changes in the revised manuscript now make our paper suitable for publication.

Best regards,

Vincent Dochez

Response to Reviewer 2:

In developed countries, endometrial cancer is currently the most common malignant neoplasm of female genital organs.

Histological verification is a widely recognized diagnostic method aimed at the diagnosis of this disease. In order to avoid aggressive invasive diagnostics and unjustified surgical intervention, other methods of early diagnosis of this neoplasm are sought.

At present, there is no biological marker used in current practice for the diagnosis of endometrial cancer.

The aim of the study submitted for review was prepare a protocol for a multicenter prospective study to evaluate the value of the HE4 marker in patients with postmenopausal bleeding for the diagnosis of endometrial cancer. In this study, the authors chose to evaluate the sensitivity of the HE4 marker as a priority.

The reviewed work fits perfectly into the current of research on this subject in leading oncology centers in the world. The proposed research methods do not raise any objections

Thank you for your synthesis and your comments.

Round 2

Reviewer 1 Report

Accept the revised version of the manuscript